D-GENIES: dot plot large genomes in an interactive, efficient and simple way

Cabanettes Floréal
http://orcid.org/0000-0001-7126-5477 Klopp Christophe christophe.klopp@toulouse.inra.fr
Plate-forme bio-informatique Genotoul, Mathématiques et Informatique Appliquées de Toulouse, INRA , Castanet Tolosan , France
Tullius Thomas
Electronic publication date: 2018 Jun 4
Publication date: 2018
Volume: 6
Electronic Location ID: e4958
Received 2018 Mar 12; Accepted 2018 May 22
Copyright: © 2018 Cabanettes and Klopp
Copyright year: 2018
Copyright holder: Cabanettes and Klopp
License: This is an open access article distributed under the terms of the Creative Commons Attribution License, which permits unrestricted use, distribution, reproduction and adaptation in any medium and for any purpose provided that it is properly attributed. For attribution, the original author(s), title, publication source (PeerJ) and either DOI or URL of the article must be cited.
License URL: https://creativecommons.org/licenses/by/4.0/

Keywords: Dot plot, Genome assessment, Interactive user interface, Large genomes

Funding: France Génomique National infrastructure (funded as part of Investissement d’Avenir Program managed by Agence Nationale pour la Recherche, contract ANR-10-INBS-0009) This work was supported by the France Génomique National infrastructure (funded as part of Investissement d’Avenir Program managed by Agence Nationale pour la Recherche, contract ANR-10-INBS-0009). The funders had no role in study design, data collection and analysis, decision to publish, or preparation of the manuscript.

==============================
Dot plots are widely used to quickly compare sequence sets. They provide a synthetic similarity overview, highlighting repetitions, breaks and inversions. Different tools have been developed to easily generated genomic alignment dot plots, but they are often limited in the input sequence size. D-GENIES is a standalone and web application performing large genome alignments using minimap2 software package and generating interactive dot plots. It enables users to sort query sequences along the reference, zoom in the plot and download several image, alignment or sequence files. D-GENIES is an easy-to-install, open-source software package (GPL) developed in Python and JavaScript. The source code is available at https://github.com/genotoul-bioinfo/dgenies and it can be tested at http://dgenies.toulouse.inra.fr/.

Introduction

Dot plots are commonly used to visually compare two sets of sequences. They present insertions, deletions, inversions or repeats in an easily understandable manner. They can represent similarity differences using variable line thickness, line forms or colors. With the increasing numbers of genome assemblies produced there is a need for simple-to-use and efficient tools to produce dot plots of large genomes.

Existing dot plot tools can be classified in two generations. The first, and oldest, comprises command line tools producing static graphics and includes among others tupple_plot (Szafranski, Jahn & Platzer, 2006) and dot-matrix (Sonnhammer & Durbin, 1995). They usually chain two processing steps, the first of which produces a match files used in the second step which renders the graphical output. They are often limited to single sequence FASTA files and do not enable any interaction with the produced graphic. Both mentioned tools are only running on Unix computers. The software packages of the second generation have been developed in java in order to be platform independent and user friendlier. They include tools such as JDotter (Brodie, Roper & Upton, 2004), Gepard (Krumsiek, Arnold & Rattei, 2007) and r2cat (Husemann & Stoye, 2009). User interaction permitted to add new dynamic features such as sequence orientation and ordering to maximize the diagonal alignment matches in order to ease visual comparison. These tools have also input sequence size limitations. For example, Gepard takes over an hour to align human chromosomes 1 vs. itself and plot the result.

A new generation of JavaScript based dot plot visualization tools emerges. One of its early members is https://dnanexus.github.io/dot/. Users have first to produce coordinates and index files before uploading them in order to render the dot plot. They can add annotations which will be displayed in the graph margins.

We present hereafter D-GENIES, an interactive, rapid and easy-to-use standalone and web application permitting to produce a complete human vs chimpanzee genome dot plot in one hour and 10 min.

Implementation

Fast dot plot computation

D-GENIES takes advantage of minimap2 (Li, 2017), one of the latest nucleic sequence alignment program which is able to map very large lowly similar multi-FASTA files. D-GENIES can only produce dot plots for nucleic sequences. In order to limit memory consumption and lower processing time, the program splits large sequence queries, such as chromosomes, in ten mega-base chunks. Processing time and memory consumption are presented in the results and discussion section hereafter.

Simple interactive user friendly interface

The Pairwise mApping Format (PAF) file is rendered in the dot plot by a JavaScript client developed with d3.js (Bostock, Ogievetsky & Heer, 2011; https://d3js.org/). To limit drawing time, only the hundred thousand largest alignments are plotted.

Both the standalone and web application are accessed through a web browser. The page top menu (Fig. 1) permits to launch a new alignment, visualize plots, browse the example gallery or documentation and contact support. To produce a dot plot, a user clicks on the Run menu item and fills three or four input boxes. A modifiable job name is automatically attributed to the dot plot. The user email address is mandatory. The application will send a message once the dot plot is rendered. The application has two modes called New alignment and Plot alignment. In New alignment mode, both, query and target FASTA files can be uploaded from the local machine or given URLs. Reference and query files can be compressed in gzip format. If no query file is provided, the reference will be aligned on itself and all trivial matches corresponding to same sequence and same positions will be removed. After hitting the submit button, the user can follow the upload and processing progression presented with different texts and progress bars. Once the job is ended, an email containing the result page link is sent to the user. The same link appears in the monitoring page. If a user has several stored results, they can be accessed using the drop down menu of the Results menu item. In Plot alignment mode, three files or links have to be provided to D-GENIES, two FASTA or FASTA index files and one alignment file in PAF or MAF format. FASTA files can be gzip compressed. In this mode, the application only renders the graphic. Once the dot plot is generated, one can download an archive containing the three files and re-upload it anytime, using the Backup file input box located on the same page, to redraw the graphic.

Figure 1 Results page view.

(A) Main menu to navigate D-GENIES pages. (B) Reference and query sequence drop down selection boxes and button to zoom in the alignment. (C) Export menu to download image files (PNG and SVG), alignment, ordered query and unaligned query or reference FASTA files. (D) Identity color panel. (E) Match size filtering slider. (F) Identity filtering entry and check boxes. (G) Strong precision check-box. (H) Line width slider. (I) Reference and query border horizontal and vertical border line slider. (J) Query sort and unsort button. (K) Noise filtering button. (L) Similarity summary button. (M) Delete job button.

The Results page (Fig. 1), when first accessed, presents the dot plot following the FASTA files sequence order. The alignment matches are presented as colored lines. Colors correspond to similarity values binned in four groups (less than 25%, between 25% and 50%, between 50% and 75% and over 75% similarity). For colorblind users, clicking on the color scale modifies the schema. A total of six color schema are already available and others can be easily added. The graphical panel top and right margins display sequence names. Depending on the sequence and name lengths, the names will be fully or partially presented. In order to ease visualization, all successive sequences smaller than 0.2% of the total length are merged in a unique super-sequence for which the margin is grayed. The left and bottom margins show the sequence size scales.

At the top of the graphical panel, you will find, on the left, two drop down text areas and a button enabling to select query and target sequences to zoom to, and on the right the Export menu. The other way of zooming in the graphical panel is to click on a given square or to push the CRTL key while turning the mouse wheel forward to zoom in and backward to zoom out. To come back to the initial view you will click on the icon in the top right angle of the graphical panel, or press the escape key. The Export menu enables to retrieve the graphic as a PNG or SVG file, suited for publication, the PAF match file and the association table which links each query with the corresponding reference sequence, as well as the ordered query FASTA file. The unaligned query and reference sequences as well as the query sequences reorganized along the reference can also be retrieved using this menu.

On the graphic right (Fig. 1), you will have access to several buttons, sliders and input boxes enabling to change color schema, filter matches on their similarity and size or because they are seen as noise. They also enable to modify match or border sizes as well as to sort query sequences relatively to the reference. A match is considered noise if its size is small and its size frequency is quite high. Therefore, we group matches by size bins, the number of bins corresponds to one tenth of the number of alignments, the bins are scanned in increasing size order to find the most represented one and from this one the one corresponding to 1% of its count is searched. All the alignments in bins smaller in size than this one are considered noise. When checked, the strong precision check-box reduces match borders removing small matches from the graphical panel, often showing gaps between non contiguous matches. The delete job button located at the bottom right of the diagram can be used to discard obsolete results.

If after sorting, the query sequence orientation does not not correspond to your expectation, it can be changed by right clicking in the graphic and selecting Reverse query. Right clicking enables also to export the complete graphic in PNG or SVG format.

To ease dot plot comparison, clicking the Summary button generates a bar graph presenting the reference similarity profile (Fig. 2), meaning the sums of match projections on the reference per similarity category divided by the total reference length. This graph is produced after sorting the query along the reference, removing included matches and noise filtering; result not shown on the graphical panel. It gives a realistic view of the overall reference and query similarity which is often not very precisely measured through visual inspection.

Figure 2 Example of identity summary.

All these features are documented in the Documentation menu item of the main menu. The Gallery menu item give access to several examples also presented in the “processing time” section of this article.

Tools and alignment file formats extensions

Adding a new aligner to D-GENIES is easy. Edit the tools.yaml file, located in the /etc/dgenies folder on unix systems, and add tool definition lines including: executable, command line skeleton, max memory usage and number of threads. If the tool does not produce a PAF formatted file, you must add the corresponding parse function in src/dgenies/lib/parsers.py and reference it in tools.yaml. Also, set split_before to True to enable query splitting ahead of mapping. If more than one aligner is available, radio buttons will appear in the Run form to select one. Their order is defined in tools.yaml, along with a help message briefly describing each tool in the interface.

New alignment formats can also be added to the Plot alignment mode of the Run form. Just define a function in /dgenies/lib/validators.py having the same name as the input file extension. This function returns True if the format is valid, False if not. Then, add a function, with the same name, in the src/dgenies/lib/parsers.py Python file, coded to transform the input file in PAF format.

Easy standalone or web server installation

D-GENIES can be installed and run as a standalone application on Unix or MS-Windows or as a web server on Unix only. It uses the Flask framework http://flask.pocoo.org/ back-end to serve web pages and submit jobs. In standalone mode only one process should be run in a given instance. In web server mode, several processes will be run simultaneously. Three steps are time and disk space consuming when working with large genomes: file upload, data preparation and alignment. D-GENIES uses three mechanisms to ensure robustness.

When installed as a web server, it can use a computer cluster to run memory and disk intensive processes through the DRMAA layer. It also uses a local scheduler storing jobs in a MySQL database which defines process order and manages concurrency on the available cores. Because files can be large and may saturate the server, their size is tested before upload. DRMAA, MySQL parameters and maximum file size are set in the configuration file.

The file folder storing the input and output files can be cleaned using the delete job button in standalone or web server mode. A cron job deleting files having more than a given number of days can also be launched periodically in web server instances.

D-GENIES can be installed using the—pip install dgenies—command and and run with the—dgenies run—command. By default, on Unix, all data is stored in the user.dgenies folder. This can be modified in the application.properties configuration file located in the /etc/dgenies folder.

The source code can be downloaded from https://github.com/genotoul-bioinfo/dgenies.

Web portal

D-GENIES can be tested using the http://dgenies.toulouse.inra.fr/ portal which permits to process up to 3 Gb reference and query sequence FASTA files.

Results

We compared D-Genies processing time with those of Gepard and r2cat for small to medium size genomes from Ensembl (https://www.ensembl.org/), this because only minimap2 can align large ones. Table 1 presents results from tests performed with standard parameters on an eight cores Intel(R) Xeon(R) CPU E5-1630 v3 @ 3.70GHz with 32 GB RAM, using one core for the alignment, this because only minimap2 runs in multiple core mode.

Table 1 Processing times for Gepard, r2cat and D-GENIES.

Reference genome	Query sequence	Gepard	r2cat	D-Genies	
E. coli K121 (5 Mb)	E. coli O157:H74 (5.5 Mb)	18 s	5 s	0.7 s	
Asp. niger1 (35 Mb)	Asp. terreus1 (30 Mb)	3 min 25 s	45 s	5 s	
Sol. lyco.2 (100 Mb)	Sol. tub.2 (90 mb)	31 min 29 s	13 min5	4 min	
Drosophila3 (146 Mb)	Drosophila3 (146 Mb)	53 min 21 s	32 min5	3 min	
A. thaliana1 (122 Mb)	A. lyrata1 (210 Mb)	1 h 23 min	55 min5	1 min	
Notes:

1 Ensembl 38 datasets.

2 Ensembl 38 datasets, Chromosome 1.

3 Ensembl 91 datasets.

4 Ensemble 38 datasets, str sakai.

5 Time without display generation: display time is high and the interface becomes buggy (memory limit is reached).

We also tested D-GENIES on a wider set of genomes coming from the same source including this time large genomes. Table 2 presents results from tests performed on a 32 cores Intel(R) Xeon(R) CPU E5-2670 v2 @ 2.50GHz with 256GB RAM server, using four cores for the minimap2 alignments.

Table 2 D-GENIES processing time and memory consumption for Ensembl datasets.

Reference genome	Query genome	Elapsed time	Maximum RAM usage	
Human1 (3.5 Gb)	Chimpanzee1 (3.4 Gb)	67 min 14 s	36 GB	
Mouse1 (3.4 Gb)	Rat1 (3.0 Gb)	39 min 54 s	24 GB	
Cow1 (2.6 Gb)	Sheep1 (2.5 Gb)	43 min 3 s	27 GB	
A. thaliana2 (135 Mb)	A. lyrata2 (206 Mb)	1 min 4 s	2 GB	
Poplar2 (417 Mb)	Vine2 (486 Mb)	3 min 21 s	8 GB	
Brassica rapa2 (284 Mb)	Brassica rapa2 (284 Mb)	2 min 52 s	8.3 Gb	
Notes:

1 Ensembl 91 datasets.

2 Ensembl 38 datasets.

Discussion

The comparison with Gepard and r2cat shows that D-GENIES processing times are respectively from eight to 83 and 3 to 55 times shorter. More often than not the difference between D-GENIES and the two other software packages increases with genome size.

Using minimap2 enables to very quickly compare large sets of lowly similar sequences. The compared sequence sets correspond to full size large genomes which can not be processed by second generation dot plot software packages. D-GENIES uses the default minimap2 parameters.

To limit minimap2 time and memory consumption, D-GENIES implements a chunking strategy. Large sequences are split in ten mega-base chunks which are aligned individually. When chunk alignments are contiguous they are visually joined in the graphical panel. With chunking, alignment duration dropped from 9 h and 45 min to 1 h and 7 min and memory consumption from 81 to 36 Gb for human vs chimpanzee comparison.

Several examples presented in the demonstration website gallery show that minimap2 is able to align quite distant genomes: Homo and Pan, Mus and Rattus, Vitis and Populus diverged, respectively, six, 16 and 114 million years ago (http://www.timetree.org/; Kumar et al., 2017). The correspondence signal decreases when the divergence time increases. For highly divergent short sequences minimap2 will hardly output any alignment. It is possible to improve the sequence correspondence by loosening minimap2 alignment parameters in the tools.yaml file or by using another aligner able to produce paf or maf file format or by adding a new aligner to D-GENIES following the procedure presented in the << tools and alignment format extension >> section.

Conclusion

New alignment algorithms and JavaScript visualization libraries enable the development of a third generation of dot plot applications. This generation is able to process large genomes in reasonable time and provides user-friendly graphical interfaces. D-GENIES can easily be extended with new aligners and new alignment file formats. Even if it has been developed to process large genomes, it is also suited for small or medium size genomes.

We thank Céline Noirot, Léo Lamy and the Bioinfo Genotoul platform system engineers and users for there feedback and help during installation and tests as well as Damien Leroux for his precious advice concerning JavaScript and Flask.

Additional Information and Declarations

Competing Interests

Author Contributions

Data Availability

The authors declare that they have no competing interests.

Floréal Cabanettes conceived and designed the experiments, performed the experiments, analyzed the data, contributed reagents/materials/analysis tools, authored or reviewed drafts of the paper, approved the final draft.

Christophe Klopp conceived and designed the experiments, analyzed the data, contributed reagents/materials/analysis tools, prepared figures and/or tables, authored or reviewed drafts of the paper, approved the final draft.

The following information was supplied regarding data availability:

GitHub: https://github.com/genotoul-bioinfo/dgenies.

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
