# Peer review of "D-GENIES: dot plot large genomes in an interactive, efficient and simple way"

_PeerJ, doi:10.7717/peerj.4958_

## Round 0.1 · original submission · Major Revisions

All three reviewers, and I, agree that the D-GENIES aligner is a useful contribution to the field. The reviewers all suggest minor changes to the manuscript that will improve the presentation.

I am most concerned, though, about the comment of Reviewer 3 concerning the differences in dotplots produced by D-GENIES and LAST. Please address this comment in your revision, perhaps by presenting similar comparisons and discussing the reproducibility of dot plots produced by various packages.

In addition, both Reviewer 1 (comment 1) and Reviewer 3 suggest that the ability to use aligners other than minmap would be important to users. Would this be possible? Please address this suggestion in your revision.

Reviewer 1 ·

Basic reporting

The article is concise and well written except a few sentences where English could be improved (e.g. line 35).

Experimental design

The article describes a set of stand alone and web-based tools to rapidly plot interactive genome-wide dot-plots. All the scripts are provided on GitHub which should enhance reproducibility.

Validity of the findings

Dgenies tools would be beneficial for users interested in pairwise genome comparisons. Authors provide a few pre-computed alignments and a way to change the colormap for better visualization and color blind persons. The website is well-designed and easy to navigate.

Additional comments

A few points that will further improve the tool are listed below:

1. In the discussion, it is not clear how to use an aligner other than minimap2. Is it only possible to use other aligners in the standalone version? It will be useful to provide a way to upload paf files (ideally, other file types too) on the web based application to view alignments constructed by other aligners.

2. SVG output file does not seem to work on my mac computer. I tried using web browser and Adobe illustrator to open it and both give an error. It seems there is something wrong with the format specification.

3. A description of the paf file and what does each column mean in the downloaded paf file must be added in the documentation for new users.

4. Time estimates in Table 1 also include the time taken by minimap2. It is not clear if the dot-plot rendering is also faster using Dgenies compared to other available packages, or is it only due to minimap2? It should be clarified.

5. It will be also be useful to allow users to download the original minimap2 alignments to assess their quality. The original alignments of each region can also be added to the tsv file, or a separate alignment file in downloads.

6. It will be a welcome addition if authors could add documentations on how to read the genome dot-plot e.g. synteny conservations, inversions etc. for students and new users.

Reviewer 2 ·

Basic reporting

no comment

Experimental design

no comment

Validity of the findings

no comment

Additional comments

The manuscript by Cabanettes and Klopp describes D-GENIES, an interactive way to generate dot plots for large genomes. D-GENIES employs minimap2 as its backend engine for sequence alignment, and the D3.js JavaScript library for frontend data visualization. It may serve as a useful new resource for comparing genomic sequences.

* Fig.1 needs to be redone. The header (with a black background) is illegible and the main image is too small. The annotations within the image are repeated in the figure caption. Also, some of the annotations are not necessary since the button labels (e.g., "Hide noice") are already self-evident. It suffices to number a few key items in the main image, and then annotate them in the caption.

* In the introduction, the authors cite several previous work on dot plots without mentioning SynMap from CoGe (https://genomevolution.org/coge/SynMap.pl). The CoGe platform is actively maintained, and SynMap is directly relevant to the present work.

* The acronym PAF (Pairwise mApping Format) is first introduced on lines 137-8 in page 5, but referenced twice on lines 48 and 76.

* D-GENIES may not be easy to install. According to the installation instruction on the web portal, D-GENIES has numerous dependencies. Root privilege is required for installation on Linux and the webserver mode "is only available for Unix systems and will NOT work on MS Windows."

* Inconsistencies: dotplot vs dot plot; javascript vs JavaScript; Fig. 1 (bold) vs fig. 1.

Reviewer 3 ·

Basic reporting

There are numerous minor English errors, but it is mostly comprehensible. Lines 83-85 are too hard to understand, and should be
re-written.

* Abstract: "to easily generated" -> "to easily generate"
* Line 71: "At the to top" -> "At the top"
* Line 128: sourve -> source?

The paper claims it can handle "lowly similar" sequences. This must
be made more precise, e.g. human-versus-rodent?
Human-versus-marsupial? Human-versus-cabbage?

Experimental design

In my opinion, visualization is important, and sometimes underrated.
A nice, interactive dotplot viewer could be extremely useful.

The paper should clarify whether D-GENIES reads alignment endpoints
only, or also reads the locations of gaps within alignments. PAF
format does not necessarily include gap locations. If D-GENIES
assumes that an alignment is a straight line between the endpoints,
this could be inaccurate if there are gaps.

Validity of the findings

This paper has a critical flaw: there is no evidence regarding the
accuracy of the dotplots. As a test, I compared the genomes of
E. coli (NC_000913.3) and Y. pestis (NC_003143.1) using D-GENIES and
also LAST (http://last.cbrc.jp/). The two dotplots look worryingly
different. Moreover, D-GENIES claims that these alignments are mostly
< 15% identity, which seems odd because random DNA has 25% identity.

The best solution, I think, is to further decouple the visualization
from the aligner. D-GENIES should allow formats other than PAF, which
is not a standard genome alignment format. Its input form should
allow input of pre-computed alignments (e.g. widely-used ones from the
UCSC genome database).

Then, D-GENIES could be used to compare different aligners, which
would be interesting and useful.

Additional comments

* I find it very annoying that an email address is mandatory. I
managed to use a fake email address (after wasting some time), and
it works just fine.

* In the result view (fig 1), what is "strong precision"?

* Care is needed when discussing run time and memory use. Many
aligners have tunable options to allow arbitrarily low run time and
memory use, at a cost in accuracy.

---

## Round 0.2 · Minor Revisions

Thank you very much for your thoughtful revisions to your manuscript. I especially appreciate the extensions that you have made to D-GENIES to make it easy to use other aligners and to plot alignments from other sources.

In a revised manuscript, I would request that you address Reviewer 3's comment concerning the ability of D-GENIES (and minimap2) to align somewhat distantly-related genomes. This need not be extensive. A simple acknowledgement of the limitations of minimap2 in performing such alignments would be sufficient, and helpful to the reader and prospective user.

Thank you again for submitting your interesting (and useful) manuscript to PeerJ.

Reviewer 1 ·

Basic reporting

No comments.

Experimental design

No comments.

Validity of the findings

No comments.

Additional comments

The authors have addressed all my comments and I have no further comments.

Reviewer 2 ·

Basic reporting

no comment

Experimental design

no comment

Validity of the findings

no comment

Additional comments

I am satisfied with the revisions of the manuscript, except for the following minor typos/inconsistencies:

* Fig. 1 (line 69) vs. Fig 1. (lines 53, 87) and Fig 2. (line 101)
* Lines 106-7: "The Gallery menu item *give* access to several examples ..."
* Line 135: "command *and and* run with the"

Reviewer 3 ·

Basic reporting

Lines 90-93 are still too vague. How are the bin boundaries defined?
Why need they be "scanned in increasing size order" to "find the most
represented one"? This is too vague: "the one corresponding to one
percent of its count".

Experimental design

D-GENIES has a bug for reverse strands in MAF files. One way to see
this: align the human and fruit fly mitochondria at
http://lastweb.cbrc.jp/, download the MAF, and upload it to D-GENIES.

Validity of the findings

I must apologize for saying that the dotplots "look worryingly
different". I cannot now replicate this. I suspect I did not notice
that LAST's dotplot is upside-down relative to D-GENIES!

Nevertheless, my main concern remains. D-GENIES uses minimap2, a very
new aligner: it is not primarily designed for aligning
distantly-related genomes, and to my knowledge there is no public
evidence regarding its accuracy for this task. More to the point, the
D-GENIES paper mentions no such evidence.

I performed more tests, by giving somewhat distantly-related sequences
to D-GENIES. I gave it the human mitochondrial genome (NC_012920.1)
and the mitochondrial genome of

fruit fly (NC_024511.2): "Sorry, we did not find any match"
sea squirt (NC_017929.1): "Sorry, we did not find any match"
lancelet (NC_000834.1): "Sorry, we did not find any match"
sea urchin (NC_001453.1): it aligns just a tiny fragment.

In all the above cases, I get extensive alignments and dotplots at
http://lastweb.cbrc.jp/ (with default values for the "Advanced
parameters").

This suggests that minimap2 (as used by D-GENIES) cannot handle
somewhat-diverged sequences. This is not fatal for the D-GENIES
paper, but the paper must be clear and honest about this.

In a similar vein, it is unacceptable that the RESULTS section has run
time and memory usage only, with no comment on the actual results. I
tested the first example in Table 1 by giving NC_000913.3 and
NZ_CP008918.1 to D-GENIES. The resulting dotplot is very sparse,
almost empty. It is trivial to achieve high speed if you find almost
no alignments! I suggest the RESULTS should include one more figure,
showing all the dotplots (from D-GENIES, Gepard, and r2cat)
side-by-side.

Additional comments

It is not critical to address the following points.

D-GENIES is already quite user-friendly, but it would become more so
by minimizing the mandatory inputs. The authors make a good argument
for an optional email address, not a mandatory one.

When plotting a pre-made alignment, only the (PAF or MAF) alignment
file should be mandatory. The "target file" and "query file" provide
the sequence lengths, but these are present in MAF and PAF. The only
case where the target and query files provide extra information is
when some sequences participate in no alignment. In such cases, if
only an alignment file is provided, presumably D-GENIES would only
show sequences that have alignments, which seems OK.

---

## Round 0.3 · accepted · Accept

I appreciate your thoughtful response to Reviewer 3. We look forward to publishing your very interesting contribution.

#